

# Large carnivore habitat suitability modelling for Romania and associated predictions for protected areas

Bogdan Cristescu[1,2], Csaba Domokos[3], Kristine J. Teichman[4] and Scott E. Nielsen[5]

[1] Department of Biological Sciences, University of Alberta, Edmonton, Canada
[2] Department of Biological Sciences, Institute for Communities and Wildlife in Africa (iCWild), University of Cape Town, Cape Town, South Africa
[3] Milvus Group Bird and Nature Protection Association, Targu Mures, Romania
[4] Department of Biology, University of British Columbia, Okanagan, Canada
[5] Department of Renewable Resources, University of Alberta, Edmonton, Canada

Corresponding author
Bogdan Cristescu,
cristesc@ualberta.ca

## ABSTRACT

Habitat characteristics associated with species occurrences represent important baseline information for wildlife management and conservation, but have rarely been assessed for countries recently joining the EU. We used footprint tracking data and landscape characteristics in Romania to investigate the occurrence of brown bear (*Ursus arctos*), gray wolf (*Canis lupus*) and Eurasian lynx (*Lynx lynx*) and to compare model predictions between Natura 2000 and national-level protected areas (gap analysis). Wolves were more likely to occur where rugged terrain was present. Increasing proportion of forest was positively associated with occurrence of all large carnivores, but forest type (broadleaf, mixed, or conifer) generally varied with carnivore species. Areas where cultivated lands were extensive had little suitable habitat for lynx, whereas bear occurrence probability decreased with increasing proportion of built areas. Pastures were positively associated with wolf and lynx occurrence. Brown bears occurred primarily where national roads with high traffic volumes were at low density, while bears and lynx occurred at medium-high densities of communal roads that had lower traffic volumes. Based on predictions of carnivore distributions, natural areas protected in national parks were most suitable for carnivores, nature parks were less suitable, whereas EU-legislated Natura 2000 sites had the lowest probability of carnivore presence. Our spatially explicit carnivore habitat suitability predictions can be used by managers to amend borders of existing sites, delineate new protected areas, and establish corridors for ecological connectivity. To assist recovery and recolonization, management could also focus on habitat predicted to be suitable but where carnivores were not tracked.

## INTRODUCTION

Long-term persistence of many large carnivore species relies on the existence of vast natural areas of core protected habitat that act as sources for the surrounding landscape (*Noss et al., 1996*; *Soulé & Terborgh, 1999*). Identifying patterns of carnivore occurrence in relation to

the distribution of natural habitats and human land use can be used to inform protected area designation (*Carroll, Noss & Paquet, 2001*) or management (*Reed & Merenlender, 2008*). In Europe, carnivore habitat suitability has been quantified in Scandinavia (e.g., *May et al., 2008*) and central European countries (e.g., *Kobler & Adamic, 2000*; *Huck et al., 2010*). However, uncertainties over habitat suitability for species of European Community interest (e.g., large carnivores) remain prevalent in countries that have only recently joined the European Union (EU).

Many large carnivores have been extirpated from their historic habitats in Europe (*Enserink & Vogel, 2006*; *Dalerum et al., 2009*). However, carnivore decline has not been uniform and while some populations have been eradicated, others have survived and even expanded or increased (*Linnell, Swenson & Andersen, 2001*; *Chapron et al., 2014*). Currently population statuses of large carnivores vary widely among EU member states. While the brown bear (*Ursus arctos* Linnaeus), gray wolf (*Canis lupus* Linnaeus) and Eurasian lynx (*Lynx lynx* Linnaeus) are now making a come-back in the EU, Romania (which joined the EU in 2007) has historically housed large and stable populations of these species (*Boitani, 2000*; *Breitenmoser et al., 2000*; *Swenson et al., 2000*; *Van Maanen et al., 2006*).

Brown bears, wolves and lynx are protected in Romania by national and EU legislation, as well as international Conventions. Carnivore habitat in Romania is protected in EU-legislated Natura 2000 sites, national parks (IUCN category II) and nature parks (IUCN category V). Because Natura 2000 sites were designated based on expert opinion, accurate spatial information (e.g., GIS) and, to a lesser degree, by incorporating previous ecological modeling outputs, Natura 2000 sites might have higher habitat value for carnivores than national-level protected areas (national parks and nature parks). Historical decisions to designate national parks and nature parks mostly revolved around areas with little anticipated economic potential (such as rugged mountainous regions), and/or were spearheaded by scientists with interests in specific habitats or ecological communities. This latter approach to protected area designation has been common practice worldwide (*Scott et al., 2001*; *Joppa & Pfaff, 2009*). Investigating differences in habitat suitability across protected area types could help strategize the channeling of limited conservation resources to protected areas that host the best carnivore habitat. In addition, comparing carnivore occurrence from confirmed distribution records and habitat suitability analyses could help identify management needs for protected areas of certain types, thereby improving protection effectiveness.

In Romania carnivore research at the country-wide extent has focused on assessing distribution patterns from raw footprint tracking data (with outputs such as Figs. S1–S3) with little consideration of underlying habitat characteristics. Yet, carnivore habitat might exist outside the current extent of carnivore distribution, and habitat delineation could assist recovery efforts and range expansion. Only two quantitative studies have investigated potential habitat for brown bear, gray wolf and Eurasian lynx in Romania. *Salvatori (2004)* applied a Mahalanobis distance to identify environmental suitability for carnivores in the Carpathian Mountains using carnivore observation records, environmental variables and expert opinion. *Van Maanen et al. (2006)* used Marxan software to identify a network of potential protected areas for Romania, under the assumption that forest, grassland and

shrub represent prime habitat for carnivores. Subsequently to these studies, a reassessment of the occurrence of EU-listed species in Romanian protected areas was recommended (*Iojă et al., 2010*).

Using an alternative design that incorporates additional information in the modelling procedure, while also contrasting candidate models in an information theoretic approach, we (1) identify habitat characteristics associated with the occurrence of the three large carnivore species present in Romania and (2) assess whether the Natura 2000 network supersedes the national (pre-Natura 2000) protected area network in relation to carnivore habitat suitability, by inspecting the predicted values of carnivore occurrence obtained at (1). Our objectives are therefore to quantitatively select a set of habitat suitability predictors that are meaningful for the carnivore occurrence dataset; and evaluate which protected area types in the current protected area system have the highest probability of carnivore presence.

## MATERIALS & METHODS

### Study area

The study encompassed the extent of Romania. The country has varied topography, including the Carpathian Mountains, hills, plateaus and plains. Elevations range from below sea level to 2,544 m. Predominant natural land cover is forest, composed of broadleaf, mixed broadleaf-conifer and conifer forest. Dominant broadleaf tree species include oak (*Quercus* spp.), European beech (*Fagus sylvatica*), European hornbeam (*Carpinus betulus*), common ash (*Fraxinus excelsior*) and silver birch (*Betula pendula*), whereas dominant conifer tree species include silver fir (*Abies alba*), Norway spruce (*Picea abies*), European larch (*Larix decidua*) and Scots pine (*Pinus sylvestris*). Small amounts of shrub habitat occur throughout the country, most commonly in the transitional areas between forest and grassland or forest and agricultural land, as well as in abandoned agricultural areas. Moors and heathlands occur above tree line whereas natural grasslands occur at the highest elevations and in the lower elevation plains region. Human population is at moderate densities compared to other European countries, with mean density of 86.1 inhabitants/km$^2$ in 2015 (http://ec.europa.eu/eurostat/tgm/table.do?tab=table&init=1&language=en&pcode=tps00003&plugin=1), although most areas outside urban centers have relatively low human densities.

### Carnivore information

During spring of each year, the Romanian Ministry of Environment, Waters and Forest coordinates a country-wide large carnivore survey across all 2,148 Wildlife Management Units (WMUs). The mean ± SD size of a WMU is 109.7 ± 31.9 km$^2$ (range = 25.7–341.8 km$^2$). The survey involves hiking designated transects to inventory carnivore footprints encountered along the transects within each WMU. At the time the survey is carried out, most brown bears including females with cubs have already emerged from winter dens, whereas wolves and Eurasian lynx are active throughout the year. The survey protocol requires recording of date and time of tracking, with landmark locations recorded along the track (*Fundatia & ICAS, 2011*). Transect length, as well as name of river basin, are

also noted. Lengths and widths of carnivore footprints encountered are recorded in a standardized form and photographs of the footprints are taken with a ruler or tape measure for reference. Tracking data are centralized at the county level and later converted to densities for each WMU (animals per 100 km$^2$), based on raw number of tracks recorded, filtered by footprint lengths and widths to try to minimize counting the same animal more than once.

Although the stated purpose is to inventory individual carnivores within each WMU, the methodology does not produce an absolute population census because some carnivores likely move between neighboring WMUs thereby leading to double-counting. Although surveys are planned in a synchronized manner over a short period across the entire country, this is not achievable for all WMUs because of different tracking conditions and because larger WMU sizes sometimes require longer sampling periods. In addition, detectability issues also would preclude an absolute census of carnivores in each WMU. These various factors likely affect density calculations and we are in agreement with *Popescu et al. (2016)*, who also expressed concerns regarding the robustness of density estimates derived as described above. For our analyses we chose to convert density estimates to coarser scale data, i.e., carnivore occurrence (1/0) per WMU, by setting all density estimates >0 as "1", and all density estimates = 0 as "0". We used carnivore data collected in the year 2011 in the country-wide carnivore survey, which is the most recent dataset available to the public. Data were digitized from published carnivore distribution-density hardcopy maps compiled by the Romanian Forest Research and Management Institute (*Jurj & Ionescu, 2011*; Figs. S1–S3).

## Spatial environmental predictors

We used a suite of polygon and polyline vector GIS layers and a 30 m resolution digital elevation model (DEM) to generate raster grids for use in statistical modelling (Table 1). The starting spatial resolution (grain) of grid cells for GIS analyses was 1 km ×1 km. This resolution is adequate for regional level spatial modelling for large carnivore species (*Rodriguez & Delibes, 2004*; *Treves et al., 2004*; *Teichman, Cristescu & Nielsen, 2013*). To obtain raster grids relevant to carnivore home range level, for all predictor variables we calculated focal statistic mean values in GIS within rectangular moving windows that had areas specific to the study species: 8 km × 8 km window for brown bears (home range = 65 km$^2$), 11 km × 11 km window for both gray wolves (home range = 128.5 km$^2$) and Eurasian lynx (home range = 129 km$^2$). Home range sizes were based on telemetry studies of large carnivores in Romania with the average home range size used for adult bears, wolves and lynx (*Promberger, 2001*; *Promberger, 2002*; *Promberger, 2003*). We then used the moving window outputs to estimate mean values of rasters within all Romanian WMUs, which were available in polygon format.

### Habitat (Natural)

*Abiotic.* We obtained a Digital Elevation Model (DEM) from the GMES RDA project (EU-DEM, https://www.eea.europa.eu/). We resampled the DEM from 30 m × 30 m to 1 km × 1 km and estimated *TRI* (terrain ruggedness index) based on *Riley, DeGloria*
**Table 1  Variables considered in modelling large carnivore occurrence in Romania.** Data were obtained based on moving window calculations in a GIS.

| Variable | Code | Units | Data range (Bear) | Data range (Wolf) | Data range (Lynx) | Linearity | Variable justification (*potential* influence to be tested in the models) | References |
|---|---|---|---|---|---|---|---|---|
| **Habitat (Natural)** | | | | | | | | |
| Abiotic | | | | | | | | |
| Terrain ruggedness index | triXmn | Unitless (index) | 2.61–95.60 | 2.61–93.70 | 2.61–93.70 | Non-linear | Carnivores might avoid flat areas because these are more likely to be used by people. Carnivores might select intermediate ruggedness for habitat security, but avoid high ruggedness because the latter incurs high energetic costs of movement and might have lower ecosystem productivity | *Nielsen, Stenhouse & Boyce (2006)*, *May et al. (2008)*, *Bouyer et al. (2015)* |
| Biotic | | | | | | | | |
| Broadleaf forest | brdlfXmn | Unitless (proportion) | 0–0.99 | 0–0.99 | 0–0.99 | Linear | Broadleaf forest is selected by all carnivores due to high productivity for plants and ungulates | *Breitenmoser et al. (2000)*, *Bongi et al. (2008)*, *Pop et al. (2018)* |
| Mixed forest | mixedXmn | Unitless (proportion) | 0–0.91 | 0–0.88 | 0–0.88 | Linear | Mixed forest is selected by all carnivores due to high-medium productivity for plants and ungulates | *Meriggi et al. (1991)*, *Breitenmoser et al. (2000)*, *Pop et al. (2018)* |
| Conifer forest | conifXmn | Unitless (proportion) | 0–0.83 | 0–0.82 | 0–0.82 | Linear | Conifer forest is weakly selected by all carnivores due to medium-low productivity for plants and ungulates | *Meriggi et al. (1991)*, *Breitenmoser et al. (2000)*, *Pop et al. (2018)* |
| Shrub/Herbaceous | shrXmn | Unitless (proportion) | 0–0.49 | 0–0.49 | 0–0.49 | Linear | Shrub and herbaceous areas might be selected by bear and wolf but not by lynx, if the latter species is a forest specialist | *Meriggi et al. (1991)*, *Niedziałkowska et al. (2006)*, *Pop et al. (2018)* |
| **Habitat (Human)** | | | | | | | | |
| Cultivation | agricXmn | Unitless (propotion) | 0–0.98 | 0–0.98 | 0–0.98 | Non-linear | Crops and orchards provide foraging attractants to bear but high densities of cultivated land might be a deterrent to all carnivores due to lack of secure habitat. Low densities of cultivated land might be tolerated by wolf and lynx | *Meriggi et al. (1991)*, *Schadt et al. (2002)*, *Can et al. (2014)* |
| Pasture | pastXmn | Unitless (proportion) | 0–0.51 | 0–0.50 | 0–0.50 | Non-linear | Carnivores might be attracted to ungulate grazing areas, non-linearly because areas with high proportion of pasture lack secure habitat | *Meriggi et al. (1991)*, *Schadt et al. (2002)*, *Roellig et al. (2014)* |
| Artificial | artifXmn | Unitless (proportion) | 0–0.53 | 0–0.51 | 0–0.51 | Linear | Built areas deter carnivores due to lack of food items and persecution by humans | *Theuerkauf et al. (2003)*, *Niedziałkowska et al. (2006)*, *Pop et al. (2018)* |
| National roads | natrdXmn | km/km$^2$ (density) | 0–0.15 | 0–0.15 | 0–0.15 | Non-linear | National roads deter carnivores due to high levels of human presence/traffic. Predictability of traffic could result in non-linear effects for some carnivores that adapt to heavily roaded areas | *Niedziałkowska et al. (2006)*, *Northrup et al. (2012)*, *Zimmermann et al. (2014)* |
| County roads | courdXmn | km/km$^2$ (density) | 0.00–0.31 | 0.00–0.31 | 0.00–0.31 | Non-linear | County roads above a threshold deter carnivores due to human presence/traffic. Predictability of traffic could result in non-linear effects for some carnivores that adapt to heavily roaded areas | *Niedziałkowska et al. (2006)*, *Northrup et al. (2012)*, *Zimmermann et al. (2014)* |
| Communal roads | comrdXmn | km/km$^2$ (density) | 0.02–0.73 | 0.02–0.72 | 0.02–0.72 | Non-linear | Communal roads above a threshold might deter carnivores due to unpredictable traffic, but carnivores might use roaded areas due to ease of movement along roads and edge effects associated with high plant and ungulate productivity | *Niedziałkowska et al. (2006)*, *Northrup et al. (2012)*, *Zimmermann et al. (2014)* |

**Notes.**

Code, variable codes used in the modelling script have carnivore species-specific suffixes ("Xmn" in code is replaced with "bmn", bear; "wmn", wolf; "lmn", lynx).

*& Elliot*'s (*1999*) ruggedness model. Terrain ruggedness has been shown to drive the distribution of mammalian species including carnivores (*Nielsen, Stenhouse & Boyce, 2006*; *May et al., 2008*) largely because areas with high ruggedness have difficult access for people.

*Biotic.* Land cover data were derived from a Corine Land Cover layer that covered the country extent (*EIONET, 2013*). We merged land cover categories to obtain a classification that we considered ecologically relevant (Table S1). Land cover reclassification resulted in 4 land cover types of natural habitat that included three forest classes (*broadleaf*, *mixed*, *conifer*) and a *shrub/herbaceous* class (Table 1). Poor natural habitats assumed to be of little value to carnivores composed only 1.6% of the country's land area and were excluded from these analyses. We separated forest classes in our candidate models because of increasing productivity of habitats ranked from low productivity in *conifer*, through more productive habitats of *mixed* and *broadleaf* forest. We excluded water bodies from all analyses by masking them in a GIS.

### Habitat (Human)
Reclassification of the original Corine Land Cover layer that covered the country extent (*IONET, 2013*) resulted in 3 land cover types representative of human-modified habitat, that included *cultivation* (plant crops and orchards), *pasture* (livestock grazing areas) and *artificial* (human infrastructure). We included a *cultivation* land cover variable primarily because bears might be attracted to cultivated lands and orchards to forage on plant foods (C Domokos, 2017, unpublished data). We used *pasture* as an individual land cover class because of a hypothesized association between carnivore and ungulate distributions, given that many European wild ungulates use pastures in spring (*Linnell & Andersen, 1995*; *Godvik et al., 2009*). We used an *artificial* land cover variable because we expected human-built areas to be avoided by carnivores (e.g., *Niedziałkowska et al., 2006*).

We exported the linear vector layer of the Romanian road network from the OpenStreetMap (OSM) project database (http://www.openstreetmap.org/, Open Database License [ODbl] v1.0). We generated 3 road layers from the original layer, each containing one road category in decreasing order of traffic levels: *national*, *county* and *communal* roads (the latter include small roads used primarily by villagers, as well as forestry roads). A large body of literature shows that roads can impose major influences on carnivore occurrence (*Forman & Alexander, 1998*; *Trombulak & Frissell, 2000*; *Coffin, 2007*), but animals often respond differently to major roads compared to smaller ones (*Mace et al., 1996*; *Northrup et al., 2012*). Roads can have direct negative effects through vehicular collisions (*Huber, Kusak & Frkovic, 1998*; *Kaczensky et al., 2003*), or indirect effects that can result in habitat fragmentation (*Graves, Farley & Servheen, 2006*), altered behavior (*McLellan & Shackleton, 1988*) and increased stress levels (*Kerley et al., 2002*).

## Statistical analyses and predictions for protected areas
### Habitat suitability models
We used logistic regression (R function *glm*) to estimate presence/absence (1/0) of carnivores across Romania, where presence/absence was derived from footprint data at the WMU level. All predictor variables were standardized before including in the

modelling procedure, by subtracting their means and dividing by standard deviations (R code attached). To account for potential autocorrelation, for all models we computed *Driscoll & Kraay (1998)* standard errors that are robust to cross-sectional autocorrelation. Calculation of these standard errors (R package *sandwich*) involved setting a lag length as proposed by *Hoechle (2007)* based on *Newey & West (1994)*. The *Driscoll & Kraay (1998)* sandwich estimator was applied after logistic regression, so parameter estimates from modelling were those from conventional logistic regression. Results are reported as odds ratios, which are obtained by exponentiating the parameter estimate ($\beta$) of a given predictor. For a one unit increase in the respective predictor, odds ratios >1 indicate an increase and odds ratios <1 a decrease in the odds of carnivore occurrence.

We formulated candidate models *a priori* to reflect competing hypotheses on determinants of carnivore occurrence. We delineated 20 competing hypotheses, grouped in 5 model sub-sets (Tables S2–S4). Carnivore occurrence at the country-level was hypothesized to be potentially best explained by *Natural* habitat characteristics that included abiotic, biotic, or abiotic & biotic variables, specifically terrain morphology and natural land cover types (Model set 1); *Human-generated* habitat features, with variables including land cover types created by humans and roads (Model set 2); *Natural* (abiotic) and *Human-generated* predictors (Model set 3); *Natural* (biotic) and *Human-generated* variables (Model set 4); and *Natural* (abiotic & biotic) and *Human-generated* variable combinations (Model set 5). The same variable combinations were used in occurrence models for each carnivore species (Tables S2–S4; variable justification: Table 1).

Highly correlated variables ($r > |0.6|$) were not included in the same model structure. Predictor variables for carnivore occurrence were included in their linear as well as squared (quadratic) form when ecologically relevant. For example, we expected carnivores to avoid areas with low terrain ruggedness because these would presumably be more likely to be used by people, select areas of intermediate ruggedness for habitat security, and avoid areas of high ruggedness because such areas incur high energetic cost of movement and might have lower ecosystem productivity (Table 1).

We ranked models using Akaike's Information Criterion (AIC) and $\Delta$AIC (*Burnham & Anderson, 2002*). For each carnivore species we compared the residual deviance of the top model with the deviance of the corresponding null model and calculated the percent deviance explained as a measure of model fit. Variance inflation factors (VIFs) were calculated to identify potential collinearity in the top models for each species, as collinearity can influence model reliability. Top models were not affected by collinearity issues except between variables that were included in the same model structure in their linear and squared forms, which are expected to be collinear without affecting the reliability of regression parameter estimates. We plotted predicted probabilities for the top model for each carnivore to help interpretation of squared terms (R package *ggplot2*) and mapped the predicted values at the WMU scale.

We used $K$-fold cross validation (*Boyce et al., 2002*) to estimate predictive accuracy of the most supported occurrence models for each species (R package *boot*). This model validation technique entails withholding a portion of the occurrence data for model testing, which occurs after the model is trained with multiple partitions of the original data. $K$-fold

cross validation is suited for situations when independent datasets are not available for model validation (*Boyce et al., 2002*). We applied Huberty's heuristic rule for deciding the model training to testing ratio (*Huberty, 1994*) for each species. Based on number of predictor variables in top models, we used ratios of 80% model training to 20% testing for all carnivores.

### Habitat in protected areas

For each carnivore species and protected area type (national park, nature park, Natura 2000), we calculated the proportion of protected areas of a given type that intersected WMUs which had confirmed presence of the respective carnivore based on raw footprint data. This allowed us to explore which protected area types had proportionally more confirmed carnivore presence records in the overlapping WMUs. Polygon layers for national and nature parks were obtained from the Romanian Ministry of Environment, Waters and Forest (http://www.mmediu.ro/beta/domenii/protectia-naturii-2/arii-naturale-protejate/). To obtain a layer for Natura 2000 sites, we clipped to the extent of Romania the Natura 2000 protected areas polygon layer (v.2012), which we obtained from the European Environment Agency (https://www.eea.europa.eu/data-and-maps/data/natura-4/natura-2000-spatial-data/natura-2000-shapefile-1). Because some Natura 2000 sites were small in extent, we imposed minimum area thresholds based on the home range sizes used in focal statistical analyses (brown bears, 65 km$^2$; gray wolf, 128.5 km$^2$; Eurasian lynx, 129 km$^2$). The rationale was that even if some small sites might have suitable habitat, the spatial extent of a site has to accommodate at least one home range of a large carnivore in order for the site to be suited for carnivore conservation.

In a separate analysis also carried out for each carnivore species, we compared predicted habitat suitability values (relative probabilities of occurrence) from occurrence models across the 3 protected area types. These predicted values were extracted in GIS as 1 km $\times$ 1 km pixel values that were then averaged across all protected areas of a given type, based on spatial extents of polygon layers delineating protected areas of the respective type. Mean predicted values were contrasted between the 3 types of protected areas, with larger means being indicative of better habitat. Standard deviations were also calculated to inspect the spread of habitat suitability values between sites of a given protected area type. No statistical testing was necessary as the data represented a complete inventory of pixels in the study area extent.

GIS procedures were performed in ArcGIS v.10.0 (ESRI, USA), Q-GIS v.2.14.1 (Open Source Geospatial Foundation Project, USA) and Geospatial Modelling Environment v.0.7.2.0 (*Beyer, 2013*). Statistical procedures were carried out in R Studio Version 1.0.143 (*RStudio Team, 2016*).

## RESULTS

Brown bear habitat was predicted to occur primarily in the Central, Central-Western and Northern parts of Romania (Fig. 1). These regions coincide roughly with the Carpathian Mountains chain and their foothills. Gray wolf habitat was also predicted in these areas,

but was more widely distributed throughout the country (Fig. 2). Predicted habitat for Eurasian lynx was located in the same broad geographic areas as bear habitat (Fig. 3).

## Brown bear
### Habitat suitability models

The top ranked model for brown bear explained >62% of the deviance in bear occurrence (Table 2) and had substantial power of prediction (mean cross-validation estimate of accuracy 0.934). For 1 unit increase in proportion of mixed forest cover or proportion of conifer forest cover, the odds of bears occurring increased by 127% (Table 3). Bear occurrence appeared to have non-linear association with areas with crops and orchards, but the confidence intervals for the linear term overlapped zero. For 1 unit increase in proportion of built area, the odds of bear occurrence decreased by 35% and the probabilities of bears occurring at increasing artificial land cover approached and reached zero (Fig. 4). An increase in density of national roads by 1 unit resulted in 58% decrease in odds of bear presence at alpha level 0.90. The relationship between communal road densities and bear occurrence was overall a positive association, but strongly non-linear. The relative probability of bear presence was close to zero at communal road densities less than 0.3 $km/km^2$, had a steep increase between road densities 0.3–0.65 $km/km^2$ and still increased at densities of 0.65–0.8 $km/km^2$, but at lower slope of the fitted curve (Fig. 4).

### Habitat in protected areas

Based on raw presence data from fooprint tracking, bears occurred proportionately most often in national parks (87%), less often in nature parks (71%) and least frequently in Natura 2000 areas (50%). Based on predicted mean relative probability of occurrence values, habitat suitability for bears was on average high in national parks, lower in nature parks, and lowest in Natura 2000 sites (Fig. 5). However, there was substantial variability in habitat suitability both between and within protected area types. For example, when considering protected area types with the lowest and highest mean occurrence probabilities, some Natura 2000 sites had higher suitability than the mean suitability recorded for national parks.

## Gray Wolf
### Habitat suitability models

The most supported gray wolf occurrence model explained >62% of the deviance (Table 2) and had substantial predictive power (mean cross-validation estimate of accuracy 0.925). Wolf occurrence had a strong positive and non-linear association with rugged terrain (Table 3). Occurrence probability increased sharply between ruggedness values of 10–40 and plateaued at maximum probability level when ruggedness surpassed 60 (Fig. 4). A 1 unit increase in proportion of area covered by broadleaf forest had an associated increase of 39% in odds of wolf presence. Conifer forest increase by 1 unit yielded a 30-fold (2926%) increase in odds of wolf occurrence. In addition, for 1 unit linear increase in pasture, the odds of wolf presence also increased by 66% at alpha level 0.90.

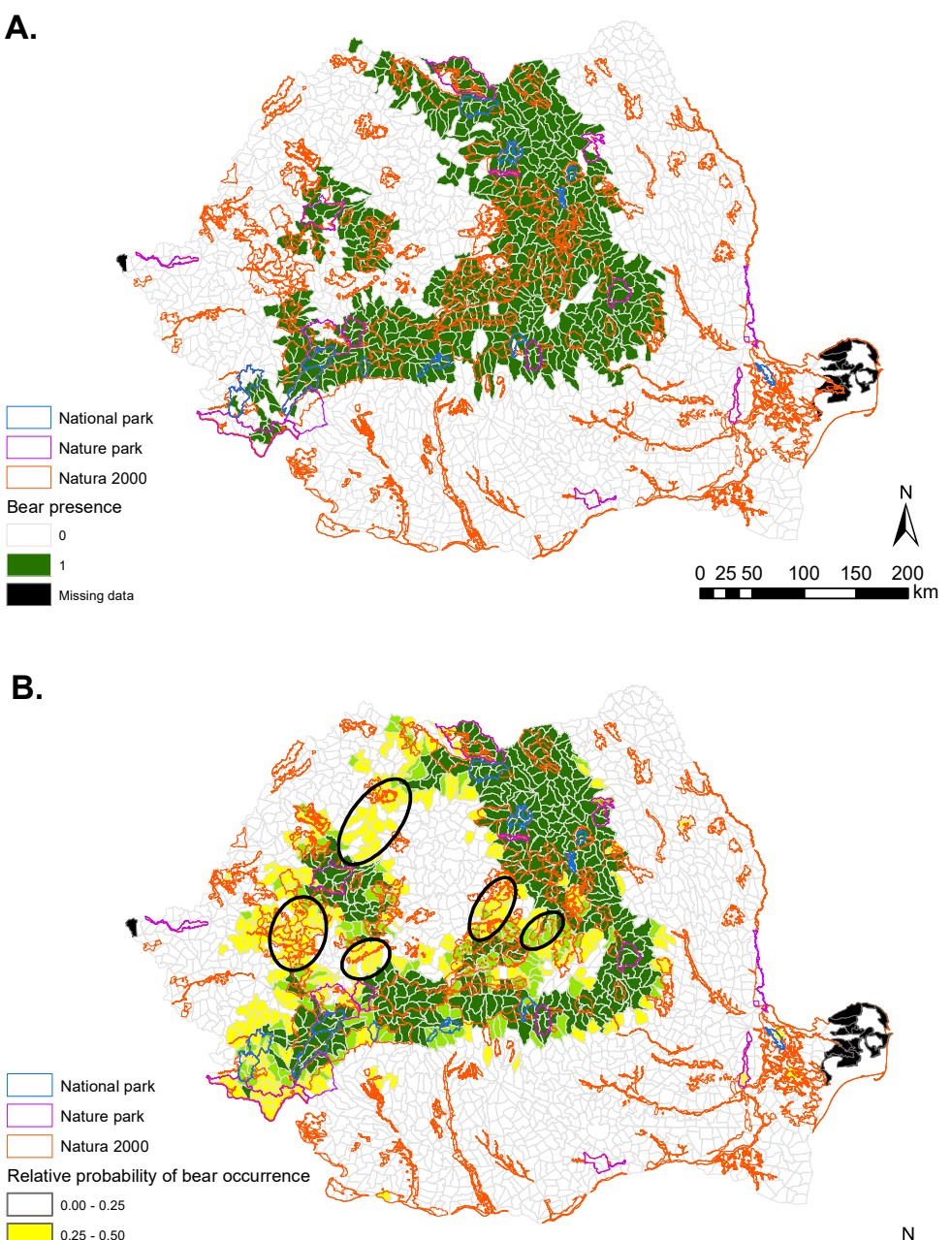

**Figure 1** **(A) Brown bear presence (1) and absence (0) based on footprint tracking in 2011 at the level of Romania's WMUs. (B) Predicted relative probabilities of brown bear occurrence based on top habitat model.** (A) Original density data mapped in *Jurj & Ionescu (2011)* were digitized and converted to 1/0 (B) Predictions refer to potential habitat, not to actual bear presence. Black ellipses provide case examples of areas where conservation efforts could focus to improve habitat suitability and establish/maintain ecological connectivity for brown bear.

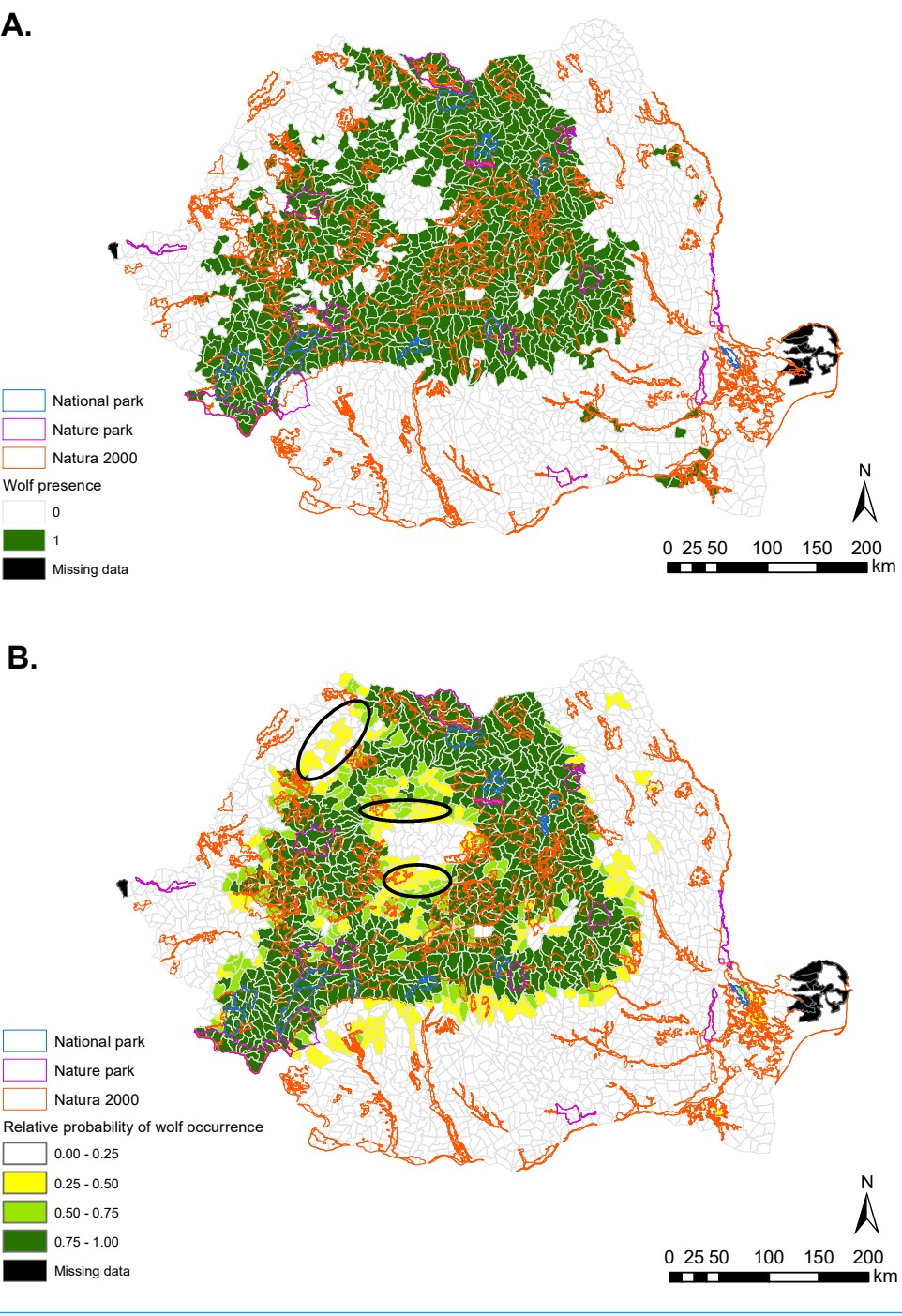

**Figure 2** **(A) Gray wolf presence (1) and absence (0) based on footprint tracking in 2011 at the level of Romania's WMUs. (B) Predicted relative probabilities of gray wolf occurrence based on top habitat model.** (A) Original density data mapped in *Jurj & Ionescu (2011)* were digitized and converted to 1/0. (B) Predictions refer to potential habitat, not to actual wolf presence. Black ellipses provide case examples of areas where conservation efforts could focus to improve habitat suitability and establish/maintain ecological connectivity for gray wolf.

**A.**

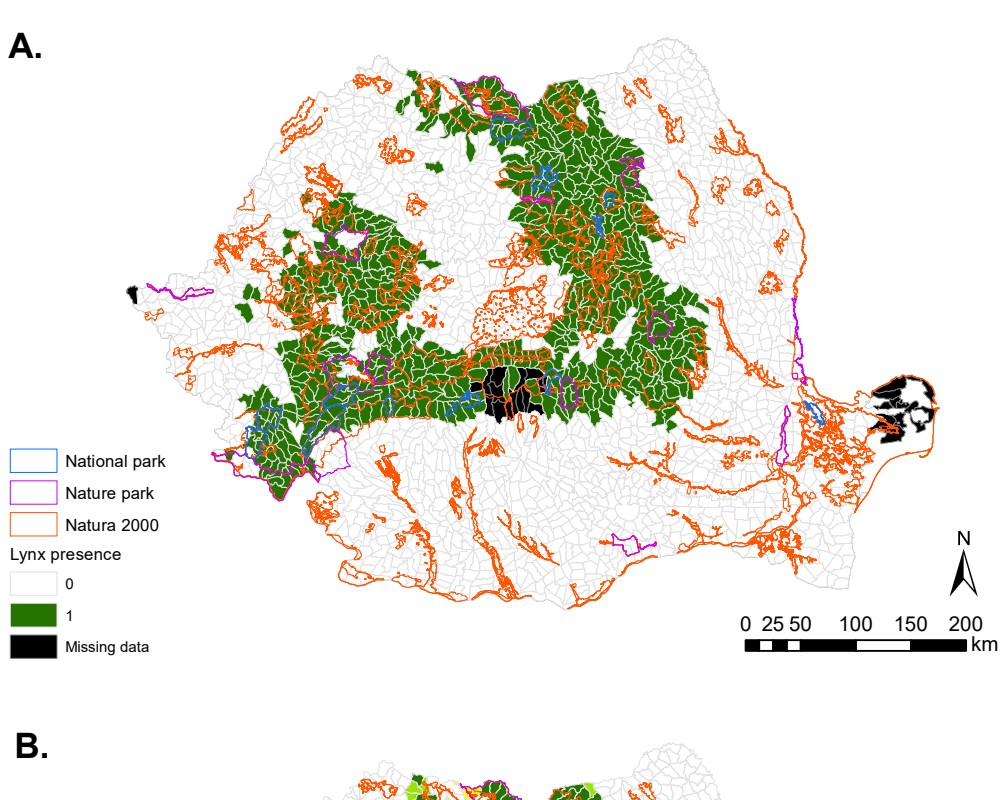

**B.**

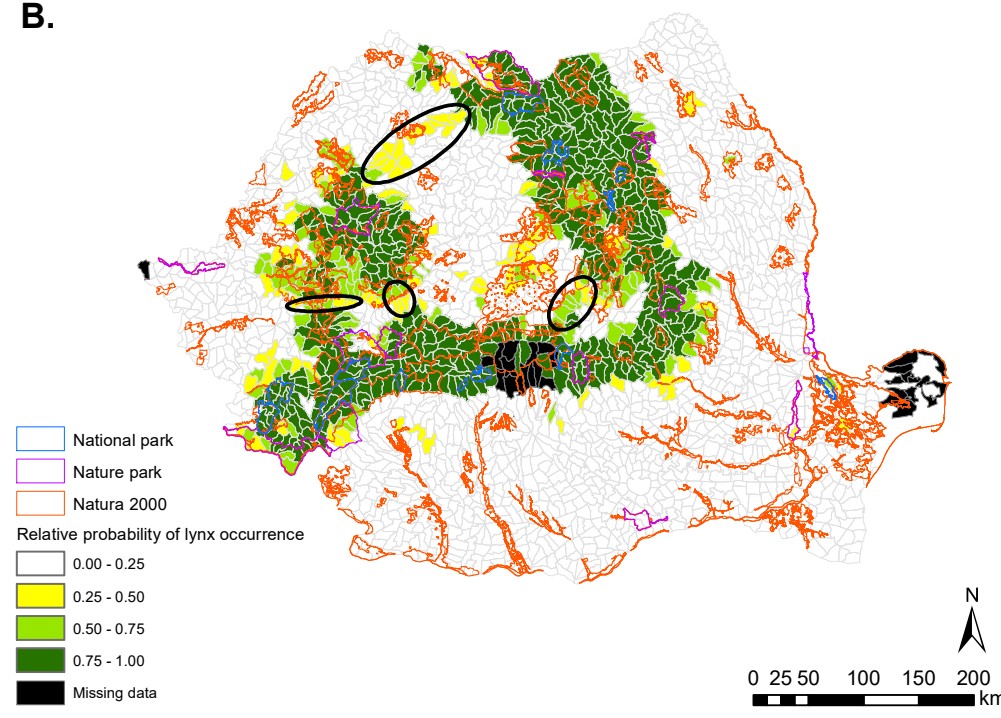

**Figure 3** **(A) Eurasian lynx presence (1) and absence (0) based on footprint tracking in 2011 at the level of Romania's WMUs. (B) Predicted relative probabilities of Eurasian lynx occurrence based on our top habitat model.** (A) Original density data mapped in *Jurj & Ionescu (2011)* were digitized and converted to 1/0. (B) Predictions refer to potential habitat, not to actual lynx presence. Black ellipses provide case examples of areas where conservation efforts could focus to improve habitat suitability and establish/maintain ecological connectivity for Eurasian lynx.

**Table 2  Top occurrence models for brown bear, gray wolf and Eurasian lynx.** Variable codes listed under Model description are provided in Table 1.

| Species | Model description | K | AIC | ΔAIC | $w_i$ | Dev. | % Dev. Expl. |
|---|---|---|---|---|---|---|---|
| Bear | mixedbmn+conifbmn+ natrdbmn+natrdbmn$^2$+ courdbmn+courdbmn$^2$+ comrdbmn+ comrdbmn$^2$+ pastbmn+pastbmn$^2$+ agricbmn+agricbmn$^2$+ artifbmn | 14 | 901.08 | 0.0 | 1.00 | 873.1 | 62.7 |
| Wolf | brdlfwmn+conifwmn+ natrdwmn+ natrdwmn$^2$+ courdwmn+courdwmn$^2$+ comrdwmn+ comrdwmn$^2$+ pastwmn+pastwmn$^2$+ artifwmn+ Striwmn+triwmn$^2$ | 14 | 1,077.1 | 0.0 | 1.00 | 1,049 | 62.4 |
| Lynx | mixedlmn+coniflmn+ natrdlmn+natrdlmn$^2$+ courdlmn+courdlmn$^2$+ comrdlmn+comrdlmn$^2$+ pastlmn+pastlmn$^2$+ agriclmn+ agriclmn$^2$+artiflmn | 14 | 692.9 | 0.0 | 0.99 | 664.9 | 72.2 |

Notes.

K, number of parameters; AIC, Akaike's Information Criterion; ΔAIC, difference in AIC between a given model and the model with the lowest AIC value in the respective model set; wi, Akaike weight; Dev., Residual Deviance; % Dev. Expl., Percentage Deviance Explained.

### Habitat in protected areas

The raw presence data from fooprint tracking showed that proportionally wolves were present primarily in national parks (87%), followed by nature parks (71%) and Natura 2000 sites (52%). Mean habitat suitability for gray wolf was also greatest for national parks, followed by nature parks and Natura 2000 sites (Fig. 5). However, similar to bears, habitat suitability for wolves varied greatly between and within protected area types, with some Natura 2000 sites having better habitat conditions than the mean suitability of national parks.

## Eurasian Lynx
### Habitat suitability models

The top model for lynx occurrence explained >72% of the deviance (Table 2) and had excellent predictive power (mean cross-validation estimate of accuracy 0.951). For 1 unit increase in proportion of mixed forest cover, the odds of lynx occurrence increased by 177% (Table 3). As proportion of crops and orchards increased by 1 unit, the odds of lynx presence decreased by 84%. Areas with pastures had a positive non-linear association with lynx occurrence. The probability of lynx occurrence increased at a steeper slope of the fit curve when proportion pasture was higher than 0.20, than when pasture was at lower values (Fig. 4). Density of communal roads was also positively non-linearly associated with

**Table 3 Parameter estimates for top brown bear, gray wolf and Eurasian lynx occurrence models.** Variable codes listed under "Variable" are provided in Table 1. Estimates for which confidence intervals did not overlap zero are given in bold.

| Variable | Bear | | | Wolf | | | Lynx | | |
|---|---|---|---|---|---|---|---|---|---|
| | $\beta$ | SE | OR | $\beta$ | SE | OR | $\beta$ | SE | OR |
| Intercept | **−3.340**[**] | **0.269** | **0.04** | −0.448 | 0.480 | 0.64 | **−3.349**[**] | **0.320** | **0.04** |
| **Habitat (Natural)** | | | | | | | | | |
| Abiotic | | | | | | | | | |
| triXmn | | | | **4.146**[**] | **0.616** | **63.18** | | | |
| triXmn$^2$ | | | | **−2.366**[**] | **0.541** | **0.09** | | | |
| Biotic | | | | | | | | | |
| brdlfXmn | | | | **0.330**[**] | **0.155** | **1.39** | | | |
| mixedXmn | **0.819**[**] | **0.178** | **2.27** | | | | **1.019**[**] | **0.341** | **2.77** |
| conifXmn | **0.818**[**] | **0.237** | **2.27** | **3.410**[**] | **1.564** | **30.26** | 0.446 | 0.470 | 1.56 |
| shrXmn | | | | | | | | | |
| **Habitat (Human)** | | | | | | | | | |
| agricXmn | 0.855 | 0.574 | 2.35 | | | | **−1.840**[**] | **0.710** | **0.16** |
| agricXmn$^2$ | **−3.681**[**] | **0.747** | **0.03** | | | | −1.376 | 0.883 | 0.25 |
| pastXmn | 0.335 | 0.235 | 1.40 | **0.505**[*] | **0.286** | **1.66** | **0.765**[**] | **0.310** | **2.15** |
| pastXmn$^2$ | 0.088 | 0.191 | 1.09 | 0.199 | 0.225 | 1.22 | **−0.664**[**] | **0.255** | **0.52** |
| artifXmn | **−0.426**[**] | **0.158** | **0.65** | −0.128 | 0.124 | 0.88 | −0.171 | 0.154 | 0.84 |
| natrdXmn | **−0.865**[*] | **0.476** | **0.42** | 0.157 | 0.341 | 1.17 | 0.022 | 0.571 | 1.02 |
| natrdXmn$^2$ | 0.798 | 0.536 | 2.22 | −0.050 | 0.336 | 0.95 | −0.163 | 0.646 | 0.85 |
| courdXmn | −0.549 | 0.513 | 0.58 | −0.339 | 0.612 | 0.71 | −0.353 | 0.773 | 0.70 |
| courdXmn$^2$ | 0.371 | 0.537 | 1.45 | 0.146 | 0.531 | 1.16 | −0.605 | 0.768 | 0.55 |
| comrdXmn | **1.658**[**] | **0.528** | **5.25** | 0.359 | 0.590 | 1.43 | **3.189**[**] | **0.717** | **24.27** |
| comrdXmn$^2$ | **−2.013**[**] | **0.501** | **0.13** | −0.782 | 0.536 | 0.46 | **−3.045**[**] | **0.721** | **0.05** |

**Notes.**

$\beta$, parameter estimate; SE, Standard error; OR, Odds Ratio.

[**]95% Confidence intervals do not overlap zero.

[*]90% Confidence intervals do not overlap zero.

lynx occurrence, with low occurrence probability at roads densities of 0–0.25 km/km$^2$, sharply increasing odds at densities of 0.25–0.50 km/km$^2$, and maximum odds at densities greater than 0.50 km/km$^2$.

### Habitat in protected areas

Similar to the other carnivores, lynx raw presence data suggested that lynx were proportionately mostly present in national parks (87%), followed by nature parks (71%) and Natura 2000 sites (42%). Mean probabilities of occurrence were also high for national parks, lower for nature parks and lowest for Natura 2000 areas (Fig. 5). As was the case for bear and wolf, habitat suitability for lynx varied between and within protected area types, with some Natura 2000 sites having better lynx habitat than the mean suitability of national parks.

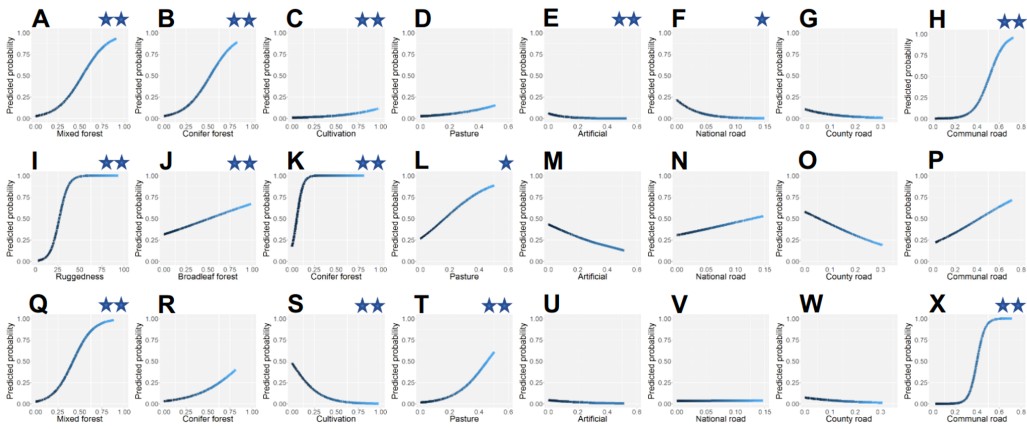

**Figure 4** **Predicted relative probabilities of brown bear (A–H), gray wolf (I–P) and Eurasian lynx (Q–X) occurrence in Romania as a function of predictor variables.** Relationships wherein confidence intervals did not overlap zero have two asterisks (95%) or one asterisk (90%).

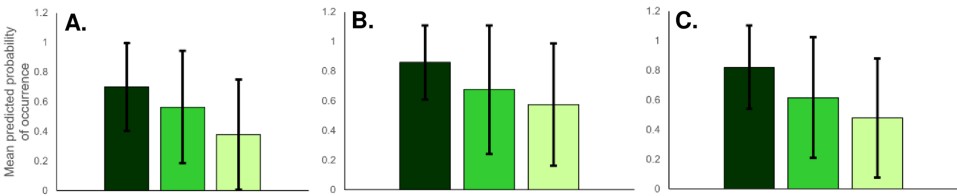

**Figure 5** **Predicted mean relative probability of occurrence of large carnivores in Romanian national parks (dark green bars), nature parks (medium green) and Natura 2000 sites (light green).** Predictions are given for (A) Brown bear; (B) Gray wolf; and (C) Eurasian lynx. Error bars represent ± SD.

## DISCUSSION

Statistical habitat models of carnivore occurrence (presence/absence) enabled us to evaluate and predict country-wide landscape responses of carnivores in Romania and habitat suitability of protected areas. The occurrences of brown bear and Eurasian lynx were best predicted by models with identical structure (Model set 4, H16) that included variables for *Natural* (biotic) and *Human-generated* habitat characteristics. The occurrence of gray wolf was best predicted by a model with variables denoting *Natural* (abiotic & biotic) and *Human-generated* features (Model set 5, H20). Our interpretations below regarding the associations between predictor variables and the response variable are based on top models, which explained most of the variation in the variable of interest (presence or absence of carnivore). This is important to emphasize because for a given predictor variable, the observed strength of its relationship with the response variable is modulated by the other predictors in the model. Basing our interpretations on outcomes of best models as identified via ranking a candidate model set, instead of on a singular global model, provided a strong foundation for inferences on carnivore habitat suitability.

Terrain ruggedness occurred in the best wolf habitat model, but not in the top bear and lynx models. Wolves showed strong selection for rugged areas, similar to wolves in Scandinavia (*May et al., 2008*). We highlight the fact that the scale of the analysis was carnivore home range level, and therefore our interpretations are home range level inferences. For example, while rugged terrain was an important predictor of wolf occurrence, this does not imply that wolves move through extremely rugged terrain, but that ruggedness is an important home range component for their persistence, possibly because it provides refuge from persecution. Throughout much of their range, wolves are one of the most highly persecuted carnivore, including in areas with legal protection (*Liberg et al., 2012*). Although evidence is limited, in Romania wolf persecution is probably widespread and has been one of the drivers for implementing applied research and conservation programs with wolf-human conflict mitigation components (Carpathian Large Carnivore Project, WOLFLIFE "Implementing best practices for the in-situ conservation of the species *Canis lupus* in the Eastern Carpathians").

Irrespective of forest type, higher proportions of forest cover were associated with increased probability of occurrence for all 3 carnivores. However, not all forest types were included in top models. Conifer forest was the only forest type that occurred in top models for all 3 carnivores, but confidence intervals overlapped zero for lynx. Brown bears can exploit vegetative foods such as berries in conifer forests even the spring following autumn berry ripening (B Cristescu, 2011, unpublished data), and they also select conifer forest for bedding (*Cristescu, Stenhouse & Boyce, 2013*). Red deer (*Cervus elaphus*) have a high proportion of conifers in their winter diets (*Gebert & Verheyden-Tixier, 2008*) and this might explain the association of wolves with conifer forests at this time of the year, given that red deer are a major prey for wolves in European ecosystems (*Okarma, 1995*). The use of mixed forests by bear and lynx has been previously documented (*Große, Kaczensky & Knauer, 2003*; *Boutros et al., 2007*) and is likely associated with food distribution, such as winter-killed ungulate carcasses (*Green, Mattson & Peek, 1997*) or berries from previous autumn for bears; and roe deer (*Capreolus capreolus*) for lynx, given that roe deer are Eurasian lynx's main prey (*Jedrzejewski et al., 1993*). The wolf's association with broadleaf forest might indicate that wolves are able to exploit ungulates which are known to use broadleaf forest in spring (*Bongi et al., 2008*).

Occurrence of lynx in areas with low proportion of cultivated land could be because human land use could displace lynx outside the cultivation season. It is unknown whether lynx in Romania show seasonal variability in occurrence on agricultural lands, or avoid these altogether. Brown bears sometimes use these areas during the timing of agricultural production (C Domokos, 2017, unpublished data), which would explain the slight association of bears with cultivated areas we found, although modelled probabilities of bear presence in these areas were low.

Wolves and lynx occurred in areas with high proportion of pastures, a pattern possibly associated with ungulate use of pastures for feeding (*Putman, 1986*; *Godvik et al., 2009*). The year-round use of pastures by wolves has been previously reported (*Meriggi et al., 1991*). Habitat edges at the forest-pasture interface were not captured in our modelling but probably present good opportunities for carnivores to hunt ungulates (*Podgórski et al.,*

*2008*). In Romania, bears also use pastures for foraging but at different times of the year than the timing of footprint surveys in the snow (*Roellig et al., 2014*).

As expected, probabilities of occurrence in relation to artificial (built) areas were low for all carnivores especially bears, for which confidence intervals for the artificial parameter estimate did not overlap zero. Built areas generally provide little natural forage, although in some areas they may provide human-originated foods that can attract carnivore and result in human-wildlife conflict situations (*Cristescu et al., 2016*). Regions with high proportion of built-up areas could also have low carnivore occurrence because of human-mediated carnivore removal, or because carnivores avoid these unsafe areas.

Avoidance of high national road density by brown bears is possibly indicative of an adverse response to vehicular traffic and comparable to grizzly bear avoidance of areas with major roads and traffic in Canada (*Gibeau et al., 2002*). In North America it is generally accepted that road development is unfavorable to grizzly bear conservation (*Nielsen, Stenhouse & Boyce, 2006*). Confidence interval overlap with zero for national road density in wolf and lynx occurrence models suggests a weaker response of these species to traffic at the current road density. This is in contrast with the avoidance of high road density by wolves in Poland (*Jedrzejewski et al., 2005*) and with extensive road networks impeding lynx movement in Germany (*Kramer-Schadt et al., 2004*). We caution that the ongoing and projected increase in the density and quality of transport infrastructure in Romania might surpass within-home range road tolerance thresholds for these species. Occurrence of brown bear and lynx in areas with medium or high communal road density, and lack of influence of communal roads on wolf occurrence, suggest that the first two species might be less responsive to human traffic on these small roads than wolf. Alternatively, lack of avoidance of areas with communal roads by all carnivores may be related to lower overall use of smaller roads by humans in spring, when many communal roads are still covered by snow, or are inaccessible because of heavy mud.

Based on both modelling predictions and raw footprint (presence-absence) data, the national park system in Romania has higher habitat suitability for bear, wolf and lynx than EU-level protected areas. In general, national parks have the strictest protective regulations and could therefore, in theory, perpetuate natural habitat types free of major human intervention, or even areas without human presence (e.g., core areas). Because national parks are strongholds for carnivores in Romania, managers of these areas should continue to enforce strict protective measures and also strive to maintain connectivity to other suitable carnivore habitat, that is ideally also formally protected.

For all carnivore species, nature parks had lower habitat suitability value than national parks, but higher mean suitability than Natura 2000 sites. However, there was substantial variability in habitat suitability both between and within protected area types, which would have remained undetected were we to explore carnivore occurrence from raw presence-absence data only. For example, identifying specific Natura 2000 sites with better habitat than some nature or national parks can assist conservation prioritization and site-level management for carnivore conservation. These differential outputs between raw and modelled carnivore occurrence illustrate the importance of incorporating quantitative techniques in assessments of protected area suitability for large carnivores. Our results

reveal that such differences are present outside protected areas also (Figs. 1–3) and are possibly indicative of carnivore range restrictions from portions of suitable habitat due to human threats. Areas with suitable carnivore habitat but where carnivores were not tracked should be targeted by management to assist in carnivore recovery and recolonization, which are possible provided favorable management (*Chapron et al., 2014*).

Nature parks have fewer restrictions on human activity than national parks, with high levels of tourism. Managers of nature parks could improve protected area suitability for carnivores and minimize the risk of carnivore-human interactions by restricting tourism activities in the best carnivore habitat. Both national and nature parks have zoning, whereas zoning is not required in the case of Natura 2000 sites. An additional key difference between the three protected area categories is that national parks and nature parks fall under Romanian legislation, whereas Natura 2000 sites are under EU legislation. While we showed that habitat suitability is greatest for national parks, the observed variability in habitat suitability for large carnivores in Romania is likely to translate differentially into true conservation effects, with EU-level legislation operating in Natura 2000 sites potentially acting as a protective framework against local or national interests that might otherwise undermine protected area effectiveness. For example, a number of nationally protected areas experience threats such as illegal logging, transport infrastructure development, mining and the construction of hydroelectric power plants. In the case of Natura 2000 sites, any potential conflictual cases between site authorities and other stakeholders, such as developers or recreationists, can be solved at the European Commission level if possibilities offered by relevant national legislation have been exhausted.

Because the Natura 2000 network has incorporated many of the pre-Natura 2000 protected areas, for a total land base that is more extensive than the national and nature parks combined (*Iojă et al., 2010*), the relative differences in habitat suitability between Natura 2000 sites and national protected areas are likely even greater than we documented. Nonetheless, the large spatial extent of the Natura 2000 network and its EU-governed legislative framework suggest that Natura 2000 sites have potential to play a significant role in conservation of large carnivore populations in Romania and likely other European countries also. Law enforcement, environmental education of local communities and viable human-wildlife conflict mitigation solutions are necessary if Natura 2000 sites and carnivore habitat in Romania in general are to achieve higher conservation value for large carnivores and species under their umbrella.

## CONCLUSIONS

We identified a variety of habitat characteristics that are associated with carnivore occurrence in Romania. Based on habitat suitability modelling, we showed that national as well as EU-legislated protected areas in Romania contain suitable habitat for large carnivores, but that the habitat values differ by carnivore species and according to protected area type. Quantitative predictions from this work could be used for border amendments of existing sites, to delineate additional protected areas, and to establish corridors for ecological connectivity (e.g., Figs. 1–3B). Even if current resources might not enable

protected area expansion, the spatial outputs from habitat suitability modelling can be used to focus management for safeguarding carnivores in areas with high habitat suitability and confirmed carnivore presence (from footprint tracking data). The results can also be used to guide the spatial prioritization of sites for implementing human-carnivore conflict mitigation programs in highly suitable carnivore habitat but where carnivore presence was not confirmed with footprint tracking; as well in areas with medium suitability for carnivores but that can facilitate connectivity between carnivore strongholds. Law enforcement and working with local communities to empower them through education and by providing tools for human-wildlife conflict mitigation can pave the way to maintaining carnivore populations, carnivore re-colonization of suitable habitat, and co-existence with humans.

## ACKNOWLEDGEMENTS

The study was part of the Milvus Group's Brown Bear Conservation and Research Program. Mark Boyce facilitated initial networking. Gábor Bóné assisted with GIS layers. Peter Damerell, Márton Atilla Kelemen, Tamás Papp and Tibor Sos provided comments on earlier drafts. The editor (Alison Boyer), Gwen Iacona and an anonymous reviewer provided excellent comments that substantially improved the manuscript.

### Funding

This work was supported by Bears in Mind (the Netherlands), Bernd Thies Foundation (Switzerland), Columbus Zoo and Aquarium (USA), EuroNatur (Germany), the Nando Peretti Foundation (Italy), Frankfurt Zoological Society (Germany), as well as the International Association for Bear Research and Management (IBA). Bogdan Cristescu was supported by an Alberta Ingenuity Fund Ph.D. scholarship, the Alberta Conservation Association and a Claude Leon Foundation postdoctoral fellowship at the University of Cape Town. The funders had no role in study design, data collection and analysis, decision to publish, or preparation of the manuscript.

### Grant Disclosures

The following grant information was disclosed by the authors:
Bears in Mind (the Netherlands).
Bernd Thies Foundation (Switzerland).
Columbus Zoo and Aquarium (USA).
EuroNatur (Germany).
Nando Peretti Foundation (Italy).
Frankfurt Zoological Society (Germany).
International Association for Bear Research and Management (IBA).
Alberta Ingenuity Fund Ph.D. scholarship.
Alberta Conservation Association and a Claude Leon Foundation postdoctoral fellowship.

## Competing Interests

Csaba Domokos is an employee of Milvus Group Bird and Nature Protection Association.

## Author Contributions

- Bogdan Cristescu conceived and designed the experiments, performed the experiments, analyzed the data, contributed reagents/materials/analysis tools, prepared figures and/or tables, authored or reviewed drafts of the paper, approved the final draft.
- Csaba Domokos conceived and designed the experiments, performed the experiments, contributed reagents/materials/analysis tools, prepared figures and/or tables, authored or reviewed drafts of the paper, approved the final draft.
- Kristine J. Teichman conceived and designed the experiments, performed the experiments, contributed reagents/materials/analysis tools, authored or reviewed drafts of the paper, approved the final draft.
- Scott E. Nielsen conceived and designed the experiments, performed the experiments, authored or reviewed drafts of the paper, approved the final draft.

## Data Availability

Raw data is available in Supplemental Material.

## Supplemental Information

Supplemental information for this article can be found online at http://dx.doi.org/10.7717/peerj.6549#supplemental-information.

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
