# Peer review of "Large carnivore habitat suitability modelling for Romania and associated predictions for protected areas"

_PeerJ, doi:10.7717/peerj.6549_

## Round 0.1 · original submission · Major Revisions

The reviewers and I generally agree that this paper examines a very interesting question with solid data. Please address the comments that the reviewers have raised regarding the raw presence-absence data for the species, the statistical analyses, and the overall framing of the question(s) addressed. I encourage you to look closely and not overstate the results that you found. Framing this as a study of the "habitat suitability for large carnivores in Romania" would be good. Please also have the manuscript edited for correct English language grammar and usage.

Reviewer 1 ·

Basic reporting

Basic reporting is solid

Experimental design

I do have several comments related to the fairly sophisticated modeling in this study.

First, it would be interesting to take a look at the raw data (simple maps of occurrence / non-occurrence used in this analysis). While investigating predictors of occurrence of large carnivores is an interesting exercise, I wonder how useful it is for the main question of the manuscript: overlap with different types of protected areas. It would really be even more interesting to run the analyses of overlap with protected areas using these raw data first. Would the authors get different results? I strongly suggest that the authors do this analysis. In case the results are not different, then the whole framework and thrust (and title) of the paper should be changed to something like ‘modeling habitat suitability for large carnivores in Romania’.
Second, more explanations are needed on the use of the Huber/White/ sandwich estimator. The authors claim that is was used to handle “heteroscedasticity and/or autocorrelation”. However, this technique does not handle spatial autocorrelation. By definition, the data used here are highly correlated spatially, especially since the land planning units used in this work can be smaller than the annual home range size of all three species. The Newey-West variance estimator is the adequate technique for handling autocorrelation (by specifying a given lag). I suggest the authors re-run their analysis using the Newey-West variance estimator and re-evaluate the findings. Moreover, the authors are running a logistic regression, so the distribution is not mis-specified; importantly, unlike Ordinary Least Square regression, heteroscedasticity is to be expected in logistic regression (you are using a binary predicted variable), and homoscedasticity of independent variables is NOT a prerequisite of implementing logistic regression.
Third, please see Hosmer, Lemeshow and Sturdivant (2013) “Applied Logistic Regression” 3rd Edition for variants that accommodate spatial autocorrelation, applicable to this analysis.

More on statistical analyses:
- were the data standardized prior to analysis? Based on Table 3, it is difficult to evaluate which of the many variables used in this analysis were most predictive (among those ). Standardization is common practice, which allows the evaluation of the relative predictive power of different variables by bringing them to the same scale. Also, the parameter estimates in Table 3 are almost useless; the authors implemented logistic regression, from which very useful information, such as odds ratios, can be extracted. Please replace these uninformative betas with odds ratios and their confidence intervals, and interpret the variables accordingly throughout the manuscript (e.g., X increase in cover type Y leads to an increase/decrease in occupancy of Z).
- For all variables for which a non-linear relation was expected, please provide hypotheses or other evidence for support for making this choice

Validity of the findings

I found that the conclusions of this study to be a bit misleading. Foremost is the assertion that different types of protected areas are “conducive” to the occurrence of large carnivore species. This is a rather vague statement and the authors imply causation, when in fact this is pure correlation. This needs to be reframed and presented for what it is: an analysis of overlap between modeled distributions of large carnivores (their habitat, actually) and protected areas. While large carnivores were likely on the species lists used to declare the new Natura 2000 sites protected areas, these new protected areas were not designed specifically to accommodate large carnivores (take for example the situation in the Western Carpathians). Few, if any, of these protected areas are actively implementing strategies for large carnivore conservation, including National Parks. Not exactly sure how these protected areas are “conducive”.
I encourage the authors to expand the ideas outlined here, and consider the limitations of their conclusions, and the potential uses of this research for management or conservation. In particular, the assertion in the Conclusions (378-383) seems far-fetched and not grounded in reality. What is the trend in declaring new protected areas in Romania? Are they still doing it? Do the authors think it is feasible to declare new PAs of sufficient size to accommodate carnivore ecology? Please bring this discussion down to the real-world situation, and make an attempt to place the findings of this work in this light.

Additional comments

This is an interesting manuscript evaluating relation between the occurrence of European large carnivores and different types of protected areas in Romania. The authors used logistic regression to model the probability of occurrence of the three carnivore species, and concluded that Natura 2000 sites overlap the most with brown bear and lynx, while National Parks overlap most with wolf distribution. Given the paucity of data on Romanian large carnivores (despite accounting for the bulk of the European populations of brown bear, wolf and lynx), the information in this manuscript has the potential to advance the conservation and management of these species in the EU.
I appreciate the modeling effort put in by the authors, and the general comments and concerns regarding the overall conclusions and the statistical analyses are aimed at improving the message and strengthen the conclusions.

A couple other comments:
- Results section: this section really needs some numbers! The authors speak in terms of greater/lower/ medium etc., but no specifics are given to what those qualitative measures mean. Interpreting odds ratios would help greatly in honing message and strengths of this modeling.
- Discussions: need to be more "down-to-earth" in terms of real-world recommendations (see details above). Calculating the odds ratios for variables in your best models should help with interpreting and discussing the findings on predictors of carnivore occurrence
- Figure 2: Are these predictions based on single-variable models, or on the full model with all the other variables at mean value? I assume is the former, given the strength of the predictions (they all reach 1!). Some of those relations are non-intuitive (national road for bear and wolf), while some of them seem insanely strong (conifer forest for wolf).

Thank you for the opportunity to review this manuscript

·

Basic reporting

The manuscript was relatively well written but there are several English usage errors that should be corrected. Examples are “have” on line 35 and “conducive of occurrence” on line 264, but I was not carefully checking so there are likely others that should be sought out and addressed.

I thought the study objective as framed by the title and abstract is very interesting, but needed a better treatment in the main text. I have suggested some ways forward in the author comments below.

Experimental design

The study design seems to be appropriate for the questions asked, but there are clarifications that are necessary for a reader to be clearly able to assess this. In particular, I was unable to discern some of the statistical model formulation details. See the comments to author below for specific comments.

Validity of the findings

The model seems appropriate for the task, and the conclusions justified. However, I would like to see some more clarification of how the models were specified and how the predictions relate to the headline objectives.

Additional comments

This study develops a predictive model of carnivore occurrence across Romania. It relates habitat and landscape features to presence absence data (footprint data) and then uses the best models as identified by AIC competition to predict where the carnivores could be present. It then aims to explore the relationship between protected area categorization and expected carnivore presence but it needs some work to better achieve this part of the study.

In general I appreciate the intent of the study and the methods that were used but the framing of the study, the details of the methods, and the descriptions of the results need to be clarified.

Major comments:
1. The study appears to be framed as looking at the distribution of potential habitat for different carnivores across the different types of protected area categories. The framing misses on two fronts.

a. First, the desired framing is not clearly laid out. For instance lines 75-77 seems to attempt to justify the framing as something to do with Natura 2000 sites but does not manage to convey to me exactly what the study is aiming to achieve. Objective 2 lays out a slightly different objective as merely coverage of carnivore habitat by the protected areas in Romania, but this doesn’t require all the background information about EU protected area types. Please carefully think about what framing is most appropriate for your study intent and then revise throughout to clarify.

b. Second, the methods and results do not convey how the predicted values were used to explore the importance of the protected area categories. Depending on which framing you choose, these sections need to be revised to clearly lay out the analysis which achieves objective 2 and explain what is found.

2. The statistical model that relates habitat and human density features to carnivore occurrence data needs to be explained more clearly.

a. From the text I could not figure out how the footprint data were converted into the ones and zeros that were present in the data table and also could not determine what the independent variables associated with each FID represented. My best guess based on the supplementary figures is that presence or absence within each WMU is used as the independent variable but that needs to be much clearer in the text.

b. I found the description of the predictor variables jumbled and confusing (Lines 158-201). This section would be improved if it was arranged according to the broad variable categories in Table 1 (Land cover, Terrain, Rivers, Roads) with each section describing the reason the predictor was included in the model, where the data came from, and what the units, scale etc of each predictor was.

c. Please explain why you thought it necessary to re-rank supported models using a measure of deviance explained? I would have thought that the AICc value would have been adequate for determining the best parsimonious models because the sample size in the competing models was the same.

d. The rectangular moving window component of the analysis (lines 195-201) needs to be explained better because based on the current text it is not apparent what was done, why, or how it relates to the predictive models.

e. Similarly, the Huber-White sandwich estimator needs to be explained and justified better. In particular, why did you choose to use this estimator as opposed to fitting a basic GLM. I went to look at the Stata code in the supplemental for clues but I could not find an explanation there either.

f. Finally, the process for deciding upon which models to specify for later AIC competition needs to be better explained. I can appreciate that you would not want to try every combination of variable because of the large number of predictors, but it is not clear to me why you chose the collection that you did, and in particular why did you choose to include non-linear combinations (any predictor squared) of predictors in the candidate model set?

Minor comments:

Line 75 and elsewhere. The term representation/ representativeness is not being used as is common in the literature. E.g., Chauvenet ALM, Kuempel CD, McGowan J, Beger M, Possingham HP (2017) Methods for calculating Protection Equality for conservation planning. PLoS ONE12(2): e0171591. https://doi.org/10.1371/journal.pone.0171591

Lines 91-94: These predictions do not match the framing of the paper or the objectives well.

Lines 143-144: This comment about individual footprints being missed implies a misunderstanding of the sampling approach. Of course sampling is going to miss some individuals!

Lines 145-155: The justification for data appropriateness needs clarification. In particular, the comment about concerns about the robustness of density estimates seems misplaced and the justification of the dataset as being appropriate because it is the only one.

Line 162: why did you exclude Natura 2000 sites with no known carnivores?

Line 217: I would like an example of when a squared term would be ecologically relevant. The provided example only includes linear terms.

Lines 230 -232: why did you calculate the VIF only for the top models? I would have calculated it for the predictors in general (in a full model) at the step where you assessed correlation of predictor variables.

Lines 243-244: this is the first mention of the different protected area groups. The reframing should make it clear earlier that this analysis will be occurring and what question it is designed to answer.

Lines 284-285: please explain why the 1% deviance rule was applied for this species.

Lines 301-334: This discussion describes lots of associations between predictor variables and the probability of occurrence. The model specification, specifically the addition of many squared terms that have unclear hypotheses as to their expected relationship with the response variable, makes this study design appropriate for identifying the best model for explaining the most variation in the variable of interest (presence or absence of carnivore) but it is not appropriate for commenting on hypotheses to do with the relationship between individual predictors and the response. This is because the observed strength of the relationship is modulated by the other predictors in the model and some of them are the squared terms with unclear meaning. Please rewrite to limit the discussion to inference that is appropriate to the model design.

Lines 335 onward: The reframed discussion needs to be focused around what the predictive models say about carnivore presence across the different types of parks if that is the framing that is chosen for the paper.

Lines 339-341: this statement cannot be made based on this model. Here you are using occurrence data to predict to protected areas. This statement could only be based on the predictions being from protected areas (i.e., protection type being included in the predicted model)

Table 1:
- Not clear what the units of habitat type are. % cover? Present in pixel?
- The units column should explain what the numbers in the data spreadsheet represent, not say whether the data is continuous or not.
- What does the linearity column mean? A non-linear predictor value would be something that was squared. This does not seem to be what you are using it to mean.

Table 2:
I don’t think the null model comparison adds anything useful

Table 3:
I would prefer to see the coefficient presented as \Beta +/- the width of the confidence interval/2

Figure 2:
Please fix the scale the x axis of these plots from 0 to 1 so they can be interpreted easily.

---

## Round 0.2 · Minor Revisions

The reviewers and I agree that this manuscript is dramatically improved. However, there are some minor edits that will further improve the clarity and scientific value of the paper. Please especially address Reviewer 1's comment about the odds ratios and Reviewer 2's point about the objectives/hypotheses. I would prefer to state these as objectives, not as formal hypotheses.

Reviewer 1 ·

Basic reporting

The article is much improved in terms of flow and narrative. The authors provide R code for all analyses. The literature cited and background information is in great shape and up to date. I have no concerns whatsoever on the Basic Reporting. Great job!

Experimental design

This work is original research and the authors performed new analyses as suggested by the previous review. The methods are rigorous and have a lot of detail (including R code and raw data). The work represents a great piece of the puzzle that is carnivore ecology in Romania. The conclusions about habitat suitability and representation of these species in Romanian protected areas are well supported.
I do not have any concerns about the validity of the analyses, but I do have a couple quick questions for the authors about the reporting of the logistic regression results. Table 3 presents coefficient estimates, yet the Results talk about odds ratios and the interpretation of carnivore-habitat relationships is also done based on odds ratios (increase in 1 unit of a habitat variable results in X% increase or decrease in occurrence). Odds rations are exp(beta). Table 3 should therefore present odds ratios, not betas. Also, sometimes the authors talk about standard deviation as the unit increase or decrease, sometimes about actual values (proportions of cover types) as the unit. What are they? I am not 100% convinced that the explanations are correct, and that the authors interpret the coefficient estimates as odds ratios. For example, at line 293 the odds are increasing by a factor of 0.82 (82%). If this was an odds ratio, then the correct explanation is that the odds are decreasing by 18% for a unit increase in cover. Another source of confusion may relate to the Intercept in those models. Depending on parameterization, the intercept could be one of the levels in a categorical variables, and the rest of the coefficients are differences from that level. As such, interpreting the odds ratio for a given variable based on that coefficient is wrong, as it is relative to the Intercept, not as a standalone value. The outcome can be a huge increase or decrease in odds ratio values bc they represent changes relative to some variable (the intercept). Maybe try to run the models w/o intercept and then obtain and interpret the odds ratios that way? This should remove the issue of comparisons relative to the intercept and showcase the real odds ratios associated with each variable.

Validity of the findings

I have no concerns about the validity of the findings. Everything is well interpreted and make sense from an ecological standpoint. The only issues (if indeed they are issues and not just my misunderstanding of the presentation of results) are with the odds ratios described above.

Additional comments

I commend the authors for a great revised draft and solid and much needed contribution to the literature on large carnivores in Romania. My comments above are meant to clarify the presentation of results, not to raise concerns about the rigor or validity of your work.
Hopefully enough science is being published on this topic that will eventually make the wildlife authorities pay attention and incorporate it into management (and move their thinking from the stone age to current times). Fingers crossed...
Thank you for the opportunity to review this work!

·

Basic reporting

This is the revised version of a paper modeling habitat suitability for large carnivores in Romania, and then examining the variation in predicted suitability across different protected area types.

The authors have done a good job incorporating many of the previously requested changes and the paper reads much more clearly and coherently. However, there are still style and clarity improvements that should be made before the paper is ready for publication. Please see the detailed comments below.

Experimental design

There are a few places where better explanation of methods would improve the manuscript. Please see the detailed comments below.

Validity of the findings

The manuscript still needs careful editing to remove statements of effect or causality that are outside the scope of inference provided by the analysis. See detailed comments below.

Additional comments

Major comments:
1. The framing of the justification for analysis of differences in suitability across different protected area types is improved, but I think it still needs work.
a. This paper essentially has two steps. 1) develop three valid predictive models that can be confidently used to say something about the expected occupancy of different regions of the landscape by the three large carnivores. 2) Use these predictions to say something about the possible variation in habitat suitability across the different protection types. In this second step it is still not clear what the “something” is that you want to explore with the predictions. I would like to see a much clearer justification for why differences in predicted suitability across protection type might be relevant for conservation or management of large carnivores in Romania in the introduction. I would then like to see a clearer discussion of how managers should interpret the trends in the results and what they could practically do with this information.
b. The inclusion of the summary of raw footprint data across the different protection types is useful, but it needs to be more carefully justified, introduced, and explained. There is a difference between where individuals of rare species currently are (potentially range restricted due to threats) and where they could potentially be (this is what your predictive model is aiming to estimate. These raw data represent the first and the predictions represent the second. They are not equivalent. But a discussion of what they do represent and the differences between them is worthwhile.

2. The description of the study aims is muddied by trying to frame the objectives as hypotheses. For hypothesis 1, it is not particularly useful to describe what predictors you expect might be useful, when what you are trying to do is just find a set of relevant predictors that provide meaningful predictions. Hypothesis 2 is more reasonable to frame in this way, but it could also be framed more clearly as a question. This relates to my comment above about needing a better set up for why you are interested in the difference in predictions across protection types.

3. I would like to see a clearer explanation about the grain of the predictor data and justification for how a predictive model that is developed using response variables that are aggregated to the WMU scale is reasonable for predicting to the 1km scale.

Minor comments:
The abstract needs work to better convey the objective and broad implications of the study and less focus on specific details that are difficult to interpret out of context. Also, the statement on lines 48-50 seems overly broad to take away from this study.

The paragraph about previous research into European carnivores (lines 73-83) currently does not add useful information to the introduction. It could be modified to explain how previous work can inform expected predictions, or why there might be expectations of variation in predicted occupancy across protection types

Line 85: I don’t think that model selection and multi-modal inference minimizes uncertainty

Lines 92-99: this text starts to justify why different protection types may have different predicted occupancy. It should be in a separate paragraph and moved to earlier in the introduction because it is informing half of your study objectives.

Lines 133-135: mention that detectability issues also would preclude an absolute census

Lines 230-234: I am unconvinced by this example why the included quadratic forms are ecologically relevant. Please clarify.

Lines 239-242: please add a citation for this statement

Line 293: This is the first mention of odds. It would be better to describe the logistic model more clearly in the methods and describe there how the odds is calculated and what it represents.

Lines 306-308: This is the first mention of assessing the raw data across the protection types. Add a section to the methods justifying and describing this.

Lines 308-310 and elsewhere: be very careful about not making strong statements such as this. You can talk about mean differences across the protection types, but there is no obvious absolute difference across the protection types because the SD bars overlap for all of them. The text at 331-332 is better.

Lines 403-405 and elsewhere. Also be careful to avoid causal language in describing possible implications of observed results. Here you could say that the patterns you are seeing could be because “human land use could displace lynx” but a low probability of occurrence says nothing about the specifics of why they might not be there.

---

## Round 0.3 · accepted · Accept

Thank you for your thoughtful revisions in response to the reviewer's feedback. Your paper is a valuable contribution to the literature.

#